# Surveillance for Hepatocellular Carcinoma in Patients with Non-Alcoholic Fatty Liver Disease: Universal or Selective?

**DOI:** 10.3390/cancers12061422

**Published:** 2020-05-31

**Authors:** Maria Corina Plaz Torres, Giorgia Bodini, Manuele Furnari, Elisa Marabotto, Patrizia Zentilin, Mario Strazzabosco, Edoardo G. Giannini

**Affiliations:** 1Gastroenterology Unit, Department of Internal Medicine, University of Genoa, IRCCS-Ospedale Policlinico San Martino, 16132 Genoa, Italy; m.plaztorres@gmail.com (M.C.P.T.); giorgia.bodini@unige.it (G.B.); manuele.furnari@unige.it (M.F.); elisa.marabotto@unige.it (E.M.); pzentilin@unige.it (P.Z.); 2Liver Center and Section of Digestive Diseases, Department of Internal Medicine, Yale University School of Medicine, New Haven, CT 06520, USA; mario.strazzabosco@yale.edu

**Keywords:** NAFLD, liver cancer, steatosis, steatohepatitis, ultrasound, alpha-fetoprotein, risk factors

## Abstract

Hepatocellular carcinoma (HCC), the most frequent primary liver cancer, is the sixth most common cancer, the fourth leading cause of cancer-related deaths worldwide, and accounts globally for about 800,000 deaths/year. Early detection of HCC is of pivotal importance as it is associated with improved survival and the ability to apply curative treatments. Chronic liver diseases, and in particular cirrhosis, are the main risk factors for HCC, but the etiology of liver disease is rapidly changing due to improvements in the prevention and treatment of HBV (Hepatitis B virus) and HCV (Hepatitis C virus) infections and to the rising incidence of the metabolic syndrome, of which non-alcoholic fatty liver (NAFLD) is a manifestation. NAFLD is now a recognized and rapidly increasing cause of cirrhosis and HCC. Indeed, the most recent guidelines for NAFLD management recommend screening for HCC in patients with established cirrhosis. Screening in NAFLD patients without cirrhosis is not recommended; however, the prevalence of HCC in this group of NAFLD patients has been reported to be as high as 38%, a proportion significantly higher than the one observed in the general population and in non-cirrhotic subjects with other causes of liver disease. Unfortunately, solid data regarding the risk stratification of patients with non-cirrhotic NAFLD who might best benefit from HCC surveillance are scarce, and specific recommendations in this field are urgently needed due to the increasing NAFLD epidemic, at least in Western countries. To further complicate matters, liver ultrasonography, which represents the current standard for HCC surveillance, has a decreased diagnostic accuracy in patients with NAFLD, and therefore disease-specific surveillance tools will be required for the early identification of HCC in this population. In this review, we summarize the most recent evidence on the epidemiology and risk factors for HCC in patients with NAFLD, with and without cirrhosis, and the evidence supporting surveillance for early HCC detection in these patients, reviewing the potential limitations of currently recommended surveillance strategies, and assessing data on the accuracy of potential new screening tools. At this stage it is difficult to propose general recommendations, and best clinical judgement should be exercised, based on the profile of risk factors specific to each patient.

## 1. Introduction

Non-alcoholic fatty liver disease (NAFLD) is considered to be the hepatic manifestation of the Metabolic Syndrome (MetS) and is closely related to obesity and insulin-resistance [1]. The spectrum of disease encompasses two phenotypes: simple fatty liver, defined by the accumulation of triglycerides in >5% of the hepatocytes, and non-alcoholic steatohepatitis (NASH), characterized by the presence of steatosis, ballooning and lobular inflammation at histology. NASH is considered the progressive form of the disease, eventually leading to fibrosis, cirrhosis and its complications, including hepatocellular carcinoma (HCC) [2].

The prevalence of NAFLD has shown an increase in the last decades in parallel with that of obesity and diabetes mellitus [3]. Indeed, around a third of the global population is estimated to have NAFLD, with a prevalence that varies widely in different geographical regions being the highest in South America (30%), followed by Asia (27%), North America (24%), Europe (23%) and Africa (13%) [4]. In recent years, due to improvements in the prevention and treatment of chronic hepatitis C (HCV) and hepatitis B virus (HBV) infections, NAFLD become—proportionally—a major cause of liver disease and also one of the leading etiologies for end-stage liver disease and HCC. In addition, the burden of disease is expected to further increase [5]. Indeed, Estes et al. forecasted that by 2030 the prevalence of NAFLD in the United States will increase by 21%, from 83.1 million (2015) to 100.9 million (2030) cases, while the prevalence of NASH will increase by 63% from 16.5 million to 27.0 million cases. Accordingly, due to both disease progression and ageing of the population, the global incidence of decompensated NAFLD-cirrhosis is estimated to increase by 168% in 2030, while that of HCC is expected to increase by 137% [5]. Thus, due to the large global prevalence of NAFLD in the general population, the burden of advanced liver disease secondary to NALFD and NAFLD-associated HCC will soon be heavily felt, not only in Western countries, but reasonably also in Eastern ones, as in Asia, together with a growing industrialization and increase in Western diet pattern and metabolic diseases, the prevalence of NAFLD has worryingly increased in the last 20 years, being nowadays around 20% in Japan, about 30% in China, and as high as 51% in Indonesia [6]. Interestingly, in comparison with other Asian countries, the incidence of HCC among NAFLD patients in Japan seems to be 4-fold (incidence 4.8 per 1000 person years in Japan versus 0.3, 0.2 and 0.5 in Taiwan, South Korea and Hong Kong, respectively) [6]. Altogether, these data suggest that ethnicity, lifestyle, and social and economic conditions might contribute to the wide variation of the prevalence, phenotype and incidence of NAFLD and NAFLD-related HCC.

Whereas the evidence for a high risk of HCC in NAFLD patient with cirrhosis is substantial and bi-annual surveillance with ultrasound (US) is universally recommended in these patients [2,7], there is increasing evidence that also NAFLD patients without cirrhosis can develop HCC, with a reported proportion of non-cirrhotic NAFLD among NAFLD-related HCC cases, as high as 50% [8,9,10,11,12,13]. However, data on this topic are scant and highly heterogeneous, as different definitions of NAFLD, NASH and stages of fibrosis have been used in different series. Furthermore, the majority of the available studies are retrospective, often including small cohorts of patients, and therefore underpowered and ultimately unable to provide solid evidence in favor or against surveillance in this population. On the other hand, a prospective analysis would require a very large number of cases and a prolonged follow-up. Therefore, the HCC risk-assessment among patients with NAFLD remains an unmet need, and it is currently unclear whether surveillance for HCC should be universally offered or only be recommended in a subset of patients carrying a clinically meaningful risk of developing primary liver cancer, where early identification of HCC is cost-effective. Decision-analysis studies have shown that in general an intervention can be considered effective when it is associated with an increase in life-expectancy of approximately 3 months, and cost-effective when it can be achieved at a cost of approximately 50,000 USD per year of life gained [14,15]. In patients with compensated cirrhosis surveillance is considered cost-effective when the annual incidence of HCC is ≥1.5%, and therefore this value is considered the threshold above which surveillance should be offered [16,17]. On the contrary, there are no published studies evaluating the cost-effectiveness of surveillance in non-cirrhotic patients. Therefore, the yearly incidence of HCC above which surveillance is cost-effective in the population of patients with chronic liver disease without liver cirrhosis is actually unknown, although it is certainly lower than in cirrhotic patients. Moreover, non-cirrhotic NAFLD patients with HCC can benefit more from early diagnosis, thus suggesting that the threshold for cost-effective surveillance should be placed at an annual HCC incidence <1.5%.

In this review, we will summarize the most recent evidence on the epidemiology of HCC in patients with NAFLD and on the risk factors for HCC in patients with NAFLD. Based on these risk factors, we will highlight the sub-populations of patients with NAFLD where HCC surveillance is indicated or should be taken into consideration. We will also discuss the potential limitations of currently recommended screening and surveillance strategies, and the accuracy of potential new screening tools.

## 2. The Global Burden of HCC in NAFLD

Hepatocellular carcinoma is the sixth most common cancer, the fourth leading cause of cancer-related deaths globally, and the most frequent primary liver cancer, accounting for around 800,000 deaths/year worldwide [18,19]. Unlike most solid cancers, patients diagnosed with HCC are frequently not eligible for curative treatments and show high mortality rates with an incidence/mortality ratio close to one. In 2015, indeed, 854,000 incident liver cancer cases and 810,000 HCC-related deaths were reported [19]. Several factors are responsible for the low applicability of curative treatments in HCC patients, including late diagnosis, presence of comorbidities, older age, decompensated liver disease and/or poor liver function the low efficacy of systemic therapies. Furthermore, decompensated liver disease often limits even the application of loco-regional treatments with palliative intents [2]. In this context, the implementation of surveillance strategies for early detection of HCC nodules is fundamental, in order to increase the probability of access to curative treatments for these patients.

Currently, chronic HBV infection accounts for 33% of liver-cancer deaths, followed by alcohol (30%), chronic HCV infection (21%), and other causes including NAFLD (16%) [19]. Whereas clear indications have been given for the surveillance of HCC in viral- and alcohol-related liver diseases, there is still controversy as to which NAFLD patients best benefit from HCC surveillance [20,21,22,23]. Patients with NAFLD have a 7-fold increased risk of HCC in comparison to the general population and, and among NAFLD patients, those with cirrhosis carry the highest risk, with an annual HCC incidence of around 10.6/1000 person-years (PY) of follow up [9,24]. Although this risk is lower than that for HCV-infected patients, the high prevalence of NAFLD raises cause for concern. In a recent analysis using steady state prevalence models, it was estimated that there are 64 million people in the US and 52 million people in Germany, France, Italy, and United Kingdom with NAFLD [25]. Furthermore, prevalent NAFLD cases are forecast to increase up to 101 million in the US by 2030, with NASH cases increasing from 1.5 million to 2.7 million [5]. Considering these estimates, the overall contribution of NAFLD to global liver cancer burden becomes comparable with that, or even greater of the other more established causes of HCC.

Consistent with the above, Baffy et al. in 2013 showed that based on the estimates of the prevalence of HCV, HBV, alcoholic liver disease and NAFLD, as well as on the estimated incidence of HCC for each etiology, NAFLD may represent the relatively major contributor to the burden of patients with HCC (Figure 1), just behind HCV [26]. Taking these estimates into account, it is not surprising that NAFLD and NASH are the underlying cause of HCC in up to 59% cases in the US [27]. Moreover, a retrospective study conducted in the United Kingdom in the period between 2000 and 2010, found NAFLD as the underlying etiology for liver disease in nearly a fifth of HCC cases and was the etiology that showed the greatest increase in prevalence, registering a 35% increase during the study period [28]. Globally, age standardized death rate due to NAFLD-related liver cancer has increased annually by 1.42% since 2012, whereas there were no increases for viral hepatitis etiologies [29]. These data indicate that NAFLD is the most rapidly growing contributor to liver-related morbidity and mortality in the Western world. Not surprisingly, NAFLD is a growing indication for liver transplantation in industrialized countries, while NASH is the fastest growing cause of HCC in candidates for liver transplantation in the United States [30,31,32,33,34].

Population attributable fraction (PAF) is the proportion of cases with disease that can be avoided by removing the underlying risk factor (for liver diseases HBV, HCV and NAFLD are examples). It is calculated using the prevalence (how common) and risk estimate (how strong) of the diseases. HCV and HBV are uncommon but strong HCC risk factors in the general population; however, their PAFs are less than that of NAFLD, as the latter is a weak but highly prevalent risk factor [26,35]. Therefore, increasing the awareness of the global burden, clinical manifestations and complications of NAFLD and implementing strategies for a correct and accurate estimation of the risk of HCC across the spectrum of disease is essential. Furthermore, surveillance should be implemented in sub-populations of patients where application of this standard of care results clinically meaningful and cost-effective.

## 3. Incidence of HCC in NAFLD-Cirrhosis

The 2016 EASL/EASD/EASO Clinical Practice Guidelines recommended HCC surveillance program with a 6-month interval US for patients with NASH-cirrhosis, and the same indication was very recently confirmed also by the AGA Clinical Practice recommendations [2,7]. The evidence for this recommendation is strong, as the association between NASH-cirrhosis and a significant increase in the risk of HCC has been extensively described. However, the evidence that surveillance for HCC in this population is above the threshold where this recommendation is cost-effective (i.e., an incident rate ≥ 1.5% per year) is not so solid [16].

Available studies show contrasting results. In fact, in a prospective global study, conducted in patients with NAFLD and compensated cirrhosis at inclusion who were followed for a median mean of 85.6 months (range: 6–297 months), the annual incidence of HCC was reported to be 0.5%, which is well below the minimum 1.5% incidence rate threshold above which surveillance is considered cost-effective [36]. In contrast, a pivotal, although retrospective, study by Ascha et al. comparing 195 patients with NASH-related cirrhosis with 315 patients with HCV-related cirrhosis without a previous history of HCC found that, within a median follow-up of 3.2 years, NASH patients had a yearly cumulative incidence of HCC of 2.6%, which was lower than the 4% yearly incidence rate observed in HCV patients, but still higher than the threshold above which HCC surveillance is considered cost-effective and therefore recommended [37,38]. Similar results were obtained in a study conducted in Japan, where the 5-year incidence of HCC among 54 patients with biopsy-proven NASH-cirrhosis without HCC at inception was 11.3% [39].

More recently, Kanwal et al. estimated the risk of HCC among patients with NAFLD seen in the United States National Veterans Health Administration system. The investigators included 296,707 NAFLD patients and 296,707 matched controls without any history of liver disease. NAFLD patients had a 7-fold increased risk of developing HCC (Hazard Ratio (HR) (95% Confidence Interval (CI)): 7.62 (5.76–10.09)) compared to controls, and among patients with NAFLD, those with cirrhosis had the highest annual incidence of HCC (10.6/1000 PY). Among this subgroup of patients, HCC risk ranged from 1.6 to 23.7 per 1000 PY based on other demographic characteristics. In more detail, men (HR: 11.05 per 1000 PY (9.83–12.39)) but not women (HR: 1.62 per 1000 PY (0.20–5.85)) had an increased risk of HCC, and a gradient towards greater risk was observed in patients aged ≥ 65 years [HR: 13.43 per 1000 PY (10.82–16.49) versus those aged < 65 years (HR: 9.74 per 1000 PY (8.46–11.17)) and in diabetics [HR: 12.36 per 1000 PY (10.67–14.24) versus non-diabetics (HR: 8.51 per 1000 PY (6.96–10.29)), with the highest risk of HCC observed in older Hispanics with cirrhosis [9]. These findings suggest that even among the cirrhotic NAFLD population, the risk of incident HCC is not the same for all patients but increases in patients with specific co-morbidities and/or demographic characteristics.

Lastly, additional evidence supporting the increased risk of HCC among NAFLD patients with cirrhosis comes from a meta-analysis by White et al. that included 61 studies and was specifically aimed at identifying the risk of HCC in the NAFLD population. The results showed that NASH-cirrhosis was consistently related to an increased risk of HCC, with a cumulative incidence ranging between 2.4% over 7 years to 12.8% over 3 years in different series [40]. Again, even among the clinic-based studies included, the cohorts of patients with biopsy-proven non-cirrhotic NAFLD or NASH, showed a negligible risk of HCC, being 0% over an average of 21 years in a Danish cohort of NAFLD subjects without significant fibrosis, while a Swedish study reported an HCC-related cumulative mortality of 3% and 6% in subjects with NAFLD and NASH, respectively, followed for two decades [41].

On the basis of this evidence, the AGA Clinical Practice Update on Screening and Surveillance for HCC in patients with NAFLD, confirms the indication to offer HCC screening to all patients with NAFLD-cirrhosis [7]. Noteworthy, the authors also recommend to enroll into screening programs those patients without a clinical diagnosis of cirrhosis but with at least two non-invasive markers suggestive for the presence of cirrhosis, such as FIB-4 (point of care) > 2.67 or Enhanced Liver Fibrosis Panel (serum-based specialized test) ≥ 11.3, and an elastography examination suggestive for cirrhosis (stiffness value ≥ 16.1 kPa) [42,43]. This recommendation is based on the evidence provided by Kanwal et al. that that a FIB-4 score ≥ 2.67 is associated with an increased risk of HCC, that is the highest in subjects with known cirrhosis (incidence rate (IR) 1.36% per year), while it is lower in patients with cirrhosis and low FIB-4 scores (IR 0.5% per year), in those without cirrhosis and FIB-4 ≥ 2.67 (IR 0.04% per year) and, as expected, the lowest in patients with low FIB-4 scores and no history of cirrhosis (0.004% per year) [9]. This finding further emphasizes the evidence that the main risk factor for HCC in NAFLD is cirrhosis, that surveillance in cirrhotic patients is justified and that non-invasive markers of advanced fibrosis, such as FIB-4 ≥ 2.67, even if associated with an increased risk of HCC, are not a reliable measure of HCC risk if used alone, since the associated incidence of HCC in patients with high FIB-4 and no other evidence suggestive for cirrhosis is negligible and well below the threshold for which surveillance would be cost-effective. Therefore, in the risk stratification of patients for HCC development, non-invasive markers need to be combined with coherent radiological or clinical parameters suggestive of cirrhosis.

More recently, a study performed among 354 Mayo Clinic patients with NAFLD-cirrhosis, showed that diabetes (HR: 4.2, 95% CI 1.2–14.2, *p* = 0.02), age and low albumin significantly predicted the development of HCC, whereas other metabolic risk factors, such as increased Body Mass Index (BMI), hyperlipidemia and hypertension, did not [44]. This finding adds more evidence to the hypothesis that even among cirrhotic patients, it is possible to identify sub-groups at higher risk requiring a stringent surveillance strategy. These data may help narrowing the population of patients with cirrhotic NAFLD in which surveillance for HCC can be offered thereby improving its cost-effectiveness. Consistent with this view, although we do agree that on the basis of the current evidence surveillance should be offered to all cirrhotic patients with NAFLD, we feel that there may be sub-populations at higher risk (e.g., older male patients with diabetes) where the risk is greater and who can be the subject of enhanced surveillance. It is, however, important to note that, patients with NAFLD-cirrhosis and those with advanced fibrosis are a minority among the patients with NAFLD, and there is evidence that HCC might also appear in NAFLD patients with milder disease. Given the prevalence of NAFLD, these cases provide a relative greater contribution to the burden of disease due to the widespread prevalence of these milder liver conditions in the general population [26,45,46,47].

## 4. Incidence of HCC in NAFLD Patients without Cirrhosis

Unlike patients with NAFLD and cirrhosis, the evidence supporting the cost-effectiveness of surveillance for HCC in NAFLD patients without cirrhosis is controversial. Indeed, while it has been repeatedly reported that, in comparison to the general population, patients with NAFLD without cirrhosis are at increased risk for HCC, the estimated incidence of HCC in non-cirrhotic NASH seems to be too low to justify screening [10,14,17]. In the VHA study by Kanwal et al., 20% of NAFLD-related HCC cases occurred in the absence of cirrhosis [8]. However, the annual incidence rate of HCC in NAFLD patients who had neither a diagnosis of cirrhosis nor a FIB-4 score ≥ 2.67 was too low to justify surveillance (0.04 per 1000 PY, 95% CI (0.04–0.05)) even if this population represented 87% of the at-risk study population [9].

As shown in the meta-analysis by White et al., most of the studies addressing the issue of HCC risk among non-cirrhotic NAFLD patients had several limitations: They were retrospective, under-powered or heterogeneous with regards to the definition of NAFLD and NASH [40]. Moreover, in the studies included in this meta-analysis, the follow-up was too short for the studied endpoint (HCC incidence), ranging from a mean follow-up of 3 years to a mean follow-up of 13 years in just one 1 natural history cohort study [48]. Lastly and noteworthy, as most of the studies were underpowered, the authors could not perform a multivariate analysis aimed at defining the risk factors for HCC in non-cirrhotic NAFLD. It is worth mentioning though, that numerous case-control and cross-sectional studies showed a higher prevalence of diabetes and obesity among patients with NAFLD or cryptogenic liver disease with metabolic comorbidities, in comparison with controls with other causes of chronic liver disease.

Additional evidence for the risk of HCC development in patients with NAFLD with no signs of advanced fibrosis or cirrhosis comes from small series [49,50,51,52]. Notably, in all these studies a highest prevalence of features of the MetS characterized patients who developed HCC as compared to patients where liver cancer did not arise, further supporting the idea that particular attention should be paid to patients with NAFLD and multiple additional risk factors. Mittal et al. also confirmed this hypothesis in a retrospective analysis of data from 1500 HCC cases from Veterans Health administration hospitals [8]. In this study, only 58% of NAFLD-related HCC cases arose in the context of cirrhosis, compared to patients with alcohol- or HCV-related HCC (72.4% and 85.6%, respectively; *p* < 0.05). Furthermore, patients with NAFLD-related HCC and MetS had a more than 5-fold higher risk of having HCC in the absence of cirrhosis compared to patients with HCV-related HCC [8]. Similarly, Dyson et al. analyzed the characteristics of 632 HCC patients referred for multidisciplinary meeting in the United Kingdom—in the period between 2000 and 2010—and showed that as much as 23% of NASH-related HCC cases were non-cirrhotic [28]. Interestingly, 31% of the patients with cirrhosis were classified as cryptogenic cirrhosis, but in this group of patients a higher prevalence of diabetes, obesity, hypertension and dyslipidemia was found, suggesting that this might have been cases of the so called “burn out NASH”, i.e., advanced cases of NASH-cirrhosis in which the histological hallmarks of NASH are no longer identifiable [28,53].

Therefore, one could argue that in the context of non-cirrhotic NAFLD, patients with metabolic comorbidities are at higher risk of HCC. However, most likely, these triggers need to interact with other factors, such as ongoing inflammation and fibrosis, to promote carcinogenesis. Indeed, a recent Swedish cohort study, including 229 patients with biopsy-proven NAFLD, showed fibrosis as the main risk factor for overall mortality, cardiovascular mortality and liver-related events, including HCC [47]. In this study, during a mean follow-up of 26.4 years (range, 6–33 years), NAFLD patients had a 7-fold increased risk of HCC (HR: 6.55, 95% CI: 2.14–20.03; *p* < 0.001) as compared with the general population, similarly to the results by Kanwal et al. [9]. Interestingly, overall mortality was not related to NASH but only to the fibrosis stage as shown by the finding that mortality did not significantly increase in patients with NAFLD Activity Score between 5 and 8 and fibrosis stage 0–2 (HR: 1.41, 95%CI: 0.97–2.06; *p* = 0.07) but increased 3-fold (HR: 3.3, 95%CI: 2.27–4.76; *p* < 0.001) in patients with fibrosis stage 3–4, irrespective of NAFLD Activity Score. Thus, in non-cirrhotic NAFLD, fibrosis stage could be a useful parameter to stratify patients in different risk categories for liver-related events, including HCC.

Consistent with these findings, Yasui et al. showed that among 87 biopsy-proven NASH patients with HCC, the risk of liver cancer tended to increase as fibrosis stage increased with a prevalence of advanced grades of fibrosis (3 or 4) in 72% of HCC cases. Another relevant finding was that, apparently and consistently with previous findings, male patients were at higher risk of HCC and tended to develop HCC at earlier stages of fibrosis [51].

Recently, another systematic review with meta-analysis aimed at determining the pooled risk of HCC in patients with NASH, both in the presence and absence of cirrhosis, was published by Stine et al. [10]. The results of this analysis confirmed the indication to screen all patients with NASH-cirrhosis for HCC. Furthermore, the overall pooled estimate from the studies included in the analysis, accounting for 3567 HCC cases in 23,059 patients, indicated that not only the overall prevalence of HCC was higher but also that non-cirrhotic NASH patients had a near 3-fold increased risk of HCC in comparison to non-cirrhotic patients with other etiologies of liver disease (Odds Ratio: 2.61, 95%CI: 1.27–5.35; *p* = 0.009). Unfortunately, data on the fibrosis stage could not be extracted, and therefore the effect of fibrosis on HCC risk could not be calculated.

Overall, these findings suggest that the risk of HCC in patients with simple fatty liver is negligible and that for patients with steatosis as the only risk factor for HCC, universal HCC surveillance may not represent a cost-effective strategy. However, there is increasing evidence that the presence of NASH and advanced fibrosis, male gender and metabolic comorbidities such as diabetes and obesity may identify sub-groups of patients for which surveillance might be cost-effective, as summarized in Table 1. Therefore, further research is needed to adequately identify those factors that are independently associated with an increased risk of HCC in patients with non-cirrhotic NAFLD. This will improve the identification of sub-categories of NAFLD patients that best benefit from surveillance and allow the implementation of treatment strategies aimed at modifying preventable risk factors for HCC development.

## 5. Additional Risk Factors for HCC in Non-Cirrhotic NAFLD

Besides NASH and the presence of advanced fibrosis or cirrhosis, other independent risk factors for HCC have been recently described, with progressively increasing evidence (Figure 2). Among them, obesity, diabetes, male gender, older age, alcohol consumption and smoking each seem to be independent risk factors for HCC. It is well known that most patients with HCC have several risk factors and the HCC risk increases almost exponentially with the number of risk factors.

Notably, insulin-resistance and obesity seem to play a pivotal role in HCC development in NAFLD, independently from the progression to cirrhosis [60,61,62,63], and might partially explain the high incidence of liver cancer in the non-cirrhotic NAFLD population. Indeed, in patients with features of the MetS as the only risk factor for liver disease HCC seems to have distinct morphological characteristics and mainly occur in the absence of significant fibrosis of the background liver [50]. Furthermore, the presence of multiple features of the MetS may act synergistically further increasing the risk of liver cancer up to 6-fold if two or more features of the MetS co-exist [64]; accordingly, there is evidence that the presence of diabetes (OR: 3.5; 95%CI: 1.3–9.2), obesity (OR: 3.5; 95%CI: 1.6–7.7), both conditions (OR: 5.2; 95%CI: 1.2–22.0) or of the MetS (OR: 2.13; 95%CI: 1.96–2.31, *p* < 0.0001), as defined by the US National Cholesterol Education Program Adult Treatment Panel III, is significantly associated with a higher risk of HCC [65,66,67].

## 6. Diabetes

The association between diabetes and HCC is dated as far as 1986 and in the past two decades increasing evidence from large cohort studies has contributed to establishing diabetes as an independent risk factor for HCC [68]. Indeed, a population-based cohort study including 153,852 diabetic patients, showed that during 1,037,417 person-years of follow-up, patients with diabetes had a 4-fold increased risk of HCC (standardized incidence rate (SIR): 4.1; 95% CI: 3.8–4.5). The risk was higher in males (SIR: 4.7; 95%CI: 4.2–5.2) than in females (SIR: 3.4; 95%CI: 2.9–3.9), and was independent from other risk factors such as alcoholism, cirrhosis and viral hepatitis [69]. Similarly, in a more recent series from the Department of Veterans Affairs in the United States, including 173,643 patients with diabetes and 650,620 controls, the risk of HCC was significantly higher among diabetics (OR 2.39 versus 0.87 per 10,000 person-years, *p* < 0.0001) [70]. A 2- to 3-fold increased risk of HCC among patients with diabetes was also reported from a population-based study including patients from the Surveillance Epidemiology and End-Results Program—Medicare linked database; notably, the higher HCC risk in diabetics persisted even after the exclusion of other major risk factors (i.e., HBV, HCV, alcoholism) in this study as well [71]. Additionally, a high prevalence of non-cirrhotic cases of HCC was found in a multi-center observational prospective study performed in Italy, comparing 145 NAFLD-related HCC cases with 611 HCV-related HCC cases [13]. Cirrhosis was present in only about 50% of NAFLD-HCC patients, in contrast to the near totality of HCV-related HCC subjects but, as expected, also in this cohort the metabolic risk factors were more often present in NAFLD patients than in controls, although an analysis aimed at assessing the causality of this finding was not performed [13]. Similar results were described in Japan in a cross-sectional multi-center study including 87 histologically proven NASH patients with HCC of whom only approximately half had cirrhosis and, notably, most patients (62%) were male, obese (62%) and had diabetes (59%) [51].

Of note, diabetes was not only found as an independent risk factor for HCC in Western series but also in studies performed in Eastern countries, with similar odds ratios [56,72,73]. Indeed, Kawamura et al. showed that in a cohort of 6508 NAFLD patients, those with diabetes had a 3-fold increased risk of HCC (HR: 3.21; 95%CI: 1.09–9.50; *p* = 0.035), independently of other risk factors [56]. Lastly, two meta-analyses have confirmed the increased incidence of HCC in diabetics independently from geographic location, alcohol consumption, history of cirrhosis, or viral hepatitis [74,75].

Overall, the data presented provide strong evidence that diabetes increases the risk for HCC regardless of the presence of cirrhosis. Furthermore, diabetes has been shown to increase the risk of hepatic decompensation as well and to be related with poorer outcome after curative treatments for HCC [75,76].

## 7. Obesity

The link between obesity and cancer has been extensively reported in studies performed in Western and Eastern countries, with an observed positive linear trend in death rates from all cancers with increasing BMI [77,78,79,80]. The underlying pathogenic mechanisms are not fully understood but a direct effect of obesity on insulin resistance and the perpetuation of a pro-inflammatory milieu associated with obesity play a key role in DNA damage. Particularly, men with a BMI higher than 35.0 have a nearly 5-fold increased relative risk (RR) of death from HCC (RR: 4.52, 95%CI 2.94–6.94) as compared with men of normal weight [80].

Even though the independent association of obesity with HCC in NAFLD is difficult to ascertain as this condition is frequently accompanied by diabetes and other features of the MetS, obesity has been accepted as an independent risk factor for liver cancer. A meta-analysis by Larsson et al studied the risk of HCC in a cohort of patients with normal BMI with that of a cohort of overweight and obese patients. The analysis showed that, compared to individuals with normal weight, those who were overweight or obese had respectively a 17% and 89% increased risk of liver cancer (RR: 1.17, 95%CI: 1.02–1.34 and 1.89, 95%CI: 1.51–2.36, respectively). Moreover, the RR for obesity was significantly higher for men (RR: 2.42, 95%CI: 1.83–3.20) than for women [81]. Further evidence shows that alternatively to BMI, other parameters reflective of body fat may be useful estimates of the risk of HCC. Indeed, in a study including 359,525 cases from the European Prospective Investigation into Cancer and Nutrition study, all anthropometric measures were positively associated with the risk of HCC [82]. Particularly, waist-to-hip and waist-to-height ratio showed the strongest association with HCC (RR comparing extreme tertiles 3.51, 95%CI: 2.09–5.87; *p* < 0.0001). Furthermore, weight gain during adulthood was positively associated with HCC as well (RR: 2.48, 95%CI: 1.49–4.13; *p* < 0.001) [82].

In line with these findings, there is evidence that the risk for HCC can arise as far from childhood in overweight children as shown by a Danish registry including 372,636 children born between 1930 and 1989. The hazard ratio (95% CI) of adult liver cancer was 1.20 (1.07–1.33) and 1.30 (1.16–1.46) per 1-unit BMI z-score increase at 7 years and 13 years of age, respectively. This means that the association between childhood BMI and HCC slightly increased with the child’s age and by the age of 13, the risk had increased to 30% for each unit increase in BMI z-score. Of note, similar associations were found in boys and girls across years of birth and after accounting for diagnoses of viral hepatitis, alcohol-related disorders, and biliary cirrhosis [83].

In the last 8 years three meta-analyses have addressed the role of obesity in the risk of primary liver cancer and have shown similar results with an overall agreement of an increased risk of HCC among overweight and obese patients, independently from other risk factors [84,85,86]. Abdominal obesity has also been related to a higher recurrence rate of HCC after radio-frequency ablation (RFA) in patients with suspected NASH [87]. Moreover, the presence of obesity has been reported to be associated with an increased risk of cancer-related mortality [80,88].

## 8. Demographic Risk Factors

Another widely accepted risk factor for HCC in all etiologies of liver disease, including NAFLD and NASH, is male gender [23]. The male-to-female ratio usually ranges between 3:1 and 5:1, but in selected series ratios between 7:1 and 9:1 have been reported [89,90]. Indeed, in all of the above mentioned studies on the incidence and risk factors for HCC in NAFLD, a significant male preponderance of HCC is reported and several studies have shown that males have an up to 4-fold increased risk of HCC as compared with females [13,23,37,56,91,92,93,94]. In the study by Yasui et al., interestingly, male subjects developed HCC at a younger age and at earlier stages of fibrosis than females [51]. This finding might be related to a protective effect of estrogen hormones against HCC development [95], to a potential favoring effect of androgens on HCC [23,96] and to the higher prevalence of MetS in the male population with NAFLD [51].

It is well known that the incidence of HCC increases with advancing age, with a peak in the seventh decade [16,23,27,40]. In line with this evidence, the vast majority of studies have shown that patients with NAFLD-related HCC are usually senior [8,13,39,49,57] and significantly older than patients with HCCs occurring on the background of other chronic liver diseases [8,50]. Moreover, patients with non-cirrhotic NAFLD-related HCC are frequently older than patients with NAFLD- related HCCs and cirrhosis [97]. Finally, two studies reported that older age was an independent risk factor for NAFLD-related HCC [11,57]. These findings might reflect a longer exposure to liver damage and, possibly, the onset of more severe degrees of liver fibrosis; unfortunately, a limitation of most studies addressing the issue of HCC in NAFLD is their retrospective design and the low rate of liver biopsies, therefore sub-analysis taking into account liver fibrosis among different age strata were impossible to perform.

Another demographic factor that has shown a significant preponderance in Western studies among NAFLD patients developing HCC is Hispanic ethnicity. Couto et al., in a cohort of 1266 liver transplant recipients between 2000 and 2010, reported that Hispanics were more likely to develop NAFLD-related HCC as compared with non-Hispanics (58% vs. 30%; *p* = 0.018) [98]. This evidence was later confirmed by an analysis from the data collected in the Surveillance Epidemiology and End-Results registries and from the liver cancer mortality data from the National Center for Health Statistics in the United States, where Hispanics aged over 50 years had a significantly higher incidence of HCC [99]. In addition, Hispanics born in the United States were found to have a higher incidence of HCC when compared to foreign-born Hispanics, suggesting that environmental, socio-economic, and cultural differences may be contributing factors as well [100]. Probably, the higher prevalence of NAFLD-related HCC among Hispanics is related to the higher prevalence of features of the MetS, NAFLD and NASH in this population, as described by earlier studies and by more recent literature where higher rates of NAFLD in Hispanics were observed, with a highest risk of HCC in older Hispanics with NAFLD-cirrhosis [8,9,100,101]. Moreover, a genetic predisposition favoring HCC onset among Hispanics has been described [102].

## 9. Genetic Predisposition

Genetic predisposition may play a role in HCC development in NAFLD patients. Indeed, homozygous carriers of the I148M variant protein of PNPLA3 have a 2-fold higher hepatic fat content than non-carriers and are at higher risk of NAFLD [102]. A study by Liu et al. demonstrated that the PNPLA3 rs738409 C→→G single nucleotide polymorphism, which encodes the I148M variant protein, had a gene-dosage effect for which an increased number of G alleles (i.e., homozygous G allele) was associated with an increased incidence of NAFLD-HCC, independently from the presence of cirrhosis, with an odds ratio as high as 12.19 when compared with the general United Kingdom population [58]. Two meta-analyses have confirmed these findings showing that cirrhotic carriers of the PNPLA3 variant protein had an increased risk of HCC (OR: 1.40, 95%CI: 1.12–1.75) [103] and that this higher HCC risk persisted even after adjusting for age, sex, and BMI [104]. More recently, Grimaudo et al. performed a prospective study on 471 patients with histologically proven NAFLD which demonstrated an independent association between PNPLA3 C > G variant and HCC (HR: 2.10, 95%CI: 1.03–4.29; *p* = 0.04 [59]. These results suggest that PNPLA3 genotype could be a useful tool to stratify the risk for HCC in the NAFLD population, of course, used in combination with other accepted risk factors.

Additional genetic variants that have been associated with an increased risk of hepatic steatosis and of progressive liver disease and fibrosis are the rs58542926 C > T variant in the TM6SF2 gene and the MBOAT7, particularly in non-cirrhotic patients [105,106] but the incidence rate, determinants of risk and direct role of these genetic variants in HCC predisposition requires further investigation. 

## 10. Lifestyle

Although the definition of NAFLD excludes the consumption of significant amounts of alcohol, the assessment of alcohol consumption is not easy to perform with precision in clinical practice as it depends on the reliability of patients’ history and is extremely difficult to ascertain in retrospective research databases. However, Ascha et al. reported that, in the context of NAFLD, any amount of alcohol consumption increases the risk of HCC and, noteworthy, even a history of social alcohol intake was associated with an increased risk of HCC as compared to non-drinkers and alcohol consumption was the strongest independent risk factor for the development of HCC (HR: 3.8, 95%CI: 1.6–8.9, *p* = 0.002) [37].

Another risk factor for HCC is tobacco smoking [107]. Even though evidence in circumscribed NAFLD cohorts is limited, an International cohort study showed that HCC incidence among NAFLD patients with advanced fibrosis who were cigarette smokers was approximately twice as higher than that of non-smokers (HR: 2.11) [93]. Furthermore, smoking accounts for 13% and 9% of HCC globally and in North America, respectively, as shown by the analysis from The Liver Cancer Pooling Project, a consortium of 14 United States-based prospective cohort studies that includes data from 1,518,741 individuals and 1423 cases of HCC. Based on these data, current smokers had increased risk of HCC (HR: 1.86, 95%CI: 1.57–2.20), while individuals who had quit for more than 30 years had a risk near equivalent to never smokers. Importantly, smoking is also associated with lower survival rates in HCC [108,109]. Therefore, complete abstinence from any alcohol as well as from tobacco smoking should be recommended in patients with NAFLD, and these recommendations have recently been endorsed by the American Gastroenterological Association [7].

Based on these studies, it is reasonable to assume that greater public health efforts are needed to implement the treatment of modifiable HCC risk factors, such as increased body weight and tobacco and alcohol consumption, in the NAFLD population with the aim of possibly reducing the risk and incidence of primary liver cancer. On the other hand, from a surveillance perspective, the development of an algorithm including some or all of the abovementioned risk factors for HCC in NAFLD (i.e., age, fibrosis stage as assessed with biopsy or non-invasive fibrosis assessment tools, presence of diabetes, BMI, presence of obesity from child age, smoking status, alcohol assumption, and possibly genetic predisposition) is needed to stratify the risk of HCC at the individual patients’ level, through the identification of clusters of non-cirrhotic NAFLD patients for which surveillance for HCC would be cost-effective. Indeed, recently, the usefulness of a risk stratification based on age and alanine aminotransferase levels (ALT) levels in NAFLD patients without cirrhosis has been suggested by a Japanese study including 18,080 patients with NAFLD and no evidence of cirrhosis [11]. In this study, the 10-year cumulative incidence of HCC was found to be highest in older patients (age > 55 years) with increased ALT (12.41%, 95%CI: 5.99–18.83). These initial findings may suggest that HCC surveillance in non-cirrhotic NAFLD could be initiated at an older age, and likely further stratified according to the presence of ALT increase. Of course, we are well aware that further investigation is needed before such a risk-stratified surveillance model can be generalized. Similarly, Ioannou et al. created an HCC risk estimation model aimed at identifying those cirrhotic NAFLD patients at higher risk of HCC with the objective of stratifying patients into risk categories and further improving the cost-effectiveness of screening [54]. The models were developed separately for NAFLD-cirrhosis and ALD-cirrhosis and included seven predictors: age, gender, diabetes, BMI, platelet count, serum albumin and aspartate aminotransferase to ALT ratio. The models showed an area under the receiver operating characteristic curve (AUROC) of 0.75 for NAFLD-cirrhosis but showed poor accuracy in the prediction of the risk at the single patient level. Nevertheless, this study showed that a risk-based screening according to this prediction model was associated with a higher standardized net benefit in comparison with an approach to screen all cirrhotic patients.

## 11. The Issue of Surveillance

To date, screening for HCC with US at 6-month intervals in NAFLD is recommended for cirrhotic patients [23,110]. However, this imaging technique presents challenges in the NAFLD population as whether the accuracy of ultrasound in patients who have a good acoustic window is adequate for the identification of early HCC, its accuracy is significantly lower in obese patients, due to the thickness of subcutaneous fat and its sound-attenuating properties, especially at increased depths, thus resulting in low imaging definition [111,112,113]. Therefore, it has been questioned whether for these subjects the use of different imaging techniques would be a more appropriate screening and surveillance tool for the early detection of HCC. Technically, Computed Tomography (CT) scan and Magnetic Resonance Imaging (MRI) have higher sensitivity than US, but these imaging techniques also have lower specificity, which would carry a higher rate of false positive results, thus triggering further investigation with repeated scans and referrals [114,115]. Moreover, CT scan and MRI are expensive and, at least for CT, expose patients to radiation risks, and therefore their routine use for the surveillance of HCC is currently not recommended, and US remains the recommended method for HCC surveillance in NAFLD patients [7,115]. However, when the quality of US images is unacceptably low, CT scan or MRI may represent a valid alternative [7,23].

To increase the detectability rate of HCC at a screening level, it has been questioned if adding alpha-fetoprotein (AFP) to routine bi-annual US would be of help. However, among patients with NAFLD-related HCC, a higher proportion of AFP non-secretors has been reported and in most HCC cases occurring on a background of fatty liver disease or steatohepatitis, the serum levels of AFP were is low [13,50], an observation often suggestive of a less aggressive tumor biology [116,117]. In fact, AFP is normal in 76%–82.5% of patients with very early HCC without viral etiology, normal aminotransferases and absence of cirrhosis, features that are commonly observed in patients with NAFLD-related HCC [117]. Therefore, the usefulness of AFP testing to increase the detectability rate of HCC during screening in NAFLD patients is universally considered to be low. Moreover, no specific recommendation for HCC screening in the NAFLD population has been developed so far. As previously reported, it is widely accepted that all NASH-cirrhosis patients should be screened for HCC, but it is also evident that the risk of HCC for these subjects is lower than that for other CLD. In this context, non-invasive accurate biomarkers are needed combined with available imaging techniques in order to improve the quality of HCC surveillance in NAFLD high-risk patients.

PIVKA-II, glypican-3 (GP3) and Squamous Cell Carcinoma Antigen -1 (SCCA-1) had been proposed as new HCC-biomarkers in a study including 19 patients with histologically proven NAFLD-related HCC and 31 patients with alcohol-related HCC [118]. The authors showed that the sensitivity and specificity of GP3 was poor (sensitivity = 68%, specificity = 46.3%), whereas the serum levels of SCCA-1 were not statistically different in patients with or without HCC. In contrast, the combination of AFP and PIVKA-II testing had higher sensitivity (94% versus 58%) than AFP alone, at a modest expense of specificity (80.5% versus 100%). However, the added benefit was only limited to the detection of more advanced HCC, therefore questioning the advantage of this test in terms of curative treatment candidacy and survival benefit.

Gray et al. performed a pilot proteomic study [119] aimed at identifying novel non-invasive markers of HCC in the setting of NAFLD with. CD5L, a novel serum protein, was identified in the sera from cirrhotic individuals with and without HCC. Even if the performance of CD5L alone as a surveillance marker for HCC was poor (AUROC = 0.495), the authors argued that, given its increasing trend in cirrhosis but not per each level or stage of fibrosis, it may be reflective of hepatocyte regeneration rather than fibrosis per se. Hence, if used in combination with other serological markers it might be useful to identify patients at higher risk of HCC, although we do feel that further research in this field is needed.

Recently, novel potential non-invasive biomarkers of HCC have been proposed by researchers in the field of metabolomic, who identified acylcarnitine species as metabolites that accumulate specifically in obesity- and NASH-related HCC tissues in mouse HCC models [120]. Specifically, Enooku et al. showed that long-chain acyl-carnitines AC14:1 and AC18:1 gradually increase with the progression of fibrosis and further increase in patients with HCC, whereas the middle-chain acylcarnitine AC5:0 exhibited the opposite trend [121].

Additionally, the possible role of serum testing of micro-RNA (mi-RNA) is under investigation, as these proteins play a role in the pathogenesis and prognosis of NAFLD. In particular, among miRNAs, the liver-specific miR-122, miR-34a and miR-16 were recently found increased in the serum of NAFLD patients and their expression was associated with liver enzymes, inflammation and fibrosis [122,123,124]. Moreover, HCC patients also present elevated plasmatic levels of miR-106b-3p, miR-101-3p and miR-1246 when compared to healthy control subjects [124]. However, despite being promising, the results of these cross-sectional studies do not currently support the use of any additional biomarker for HCC surveillance in NAFLD patients.

## 12. Conclusions

Current literature demonstrates that the risk of HCC among NAFLD patients is significantly higher than that of the general population and that in consideration of the high prevalence of the disease, NAFLD-related HCC will become a leading cause of morbidity, mortality and liver transplantation in the near future. The phenotype of HCCs emerging in the context of NAFLD seems to be distinct, as it can develop not only upon a cirrhotic liver but also at earlier stages of fibrosis. The reasons behind this are not completely understood, but probably the pathology of HCC in steatosis is unique and related to steatosis, insulin resistance and to a pro-inflammatory status supported by both diabetes and obesity and other conditions inherent to the MetS as well. However, cirrhosis remains the main risk factor and in patients with NAFLD-related cirrhosis surveillance for HCC by means of bi-annual US is recommended. For non-cirrhotic patients, the presence of advanced fibrosis seems to remain an important risk factor for the occurrence of HCC, as in patients with other etiologies of liver disease, but other co-factors such as diabetes, obesity, older age, and possibly a genetic predisposition, may further enhance the risk of HCC development. As a fact, elderly males with diabetes and obesity seem to represent the cohort of patients at highest risk of HCC among non-cirrhotic NAFLD patients, and Hispanics seem to be at an even higher risk. However, we still do not have a clear perception of how these factors may be used to stratify patients’ risk and to build a risk-score to help identify the sub-population(s) with non-cirrhotic NAFLD where surveillance for HCC may be cost-effective. The currently observed low (and mostly unknown) incidence of HCC in non-cirrhotic NAFLD patients does not justify systematic surveillance in this population but suggests that a stratification by fibrosis scores and additional clinical and biochemical markers of HCC risk are warranted (see above). Therefore, future clinical research should focus on defining clusters of non-cirrhotic NAFLD patients for which systematic surveillance would be beneficial. Optimally, study design would be prospective, international, multicenter, with large cohorts and strict definitions of steatosis and NASH. The formulation of dedicated scores and the investigation of the role of non-invasive markers of fibrosis would be extremely useful, as well as the definition, for the latter, of specific cut-offs aimed at identifying those patients at higher risk of HCC. Finally, novel non-invasive markers of HCC are needed in order to improve the detectability rate at a screening level since the current recommended imaging technique (US) for HCC screening and surveillance presents challenges in the NAFLD population and might therefore cause an under-detection of the tumor at early, curative stages.

Search strategy and selection criteria: we identified references for this review through a search of PubMed with the terms “hepatocellular carcinoma”, “steatosis”, “non-alcoholic fatty liver disease”, and “steatohepatitis” from Jan 1, 2000, to Feb 29, 2020. Only papers published in English were reviewed.

## Figures and Tables

**Figure 1 cancers-12-01422-f001:**
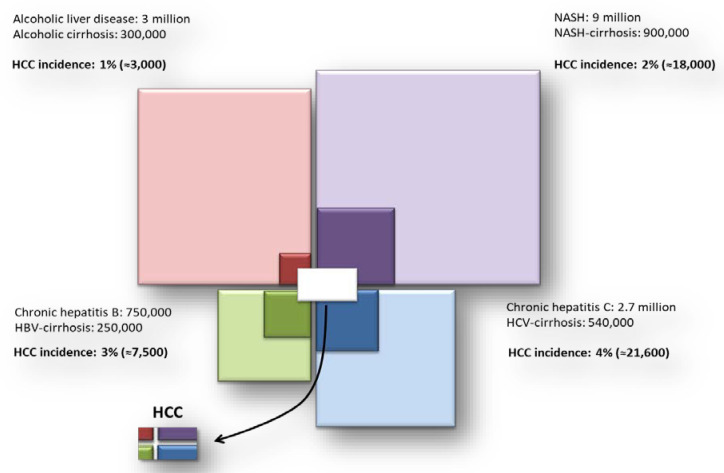
Relative contributions for HCC incidence from the most frequent causes of liver disease. HCC = hepatocellular carcinoma; NASH = non-alcoholic steato-hepatitis; HBV = hepatitis B virus; HCV = hepatitis C virus.

**Figure 2 cancers-12-01422-f002:**
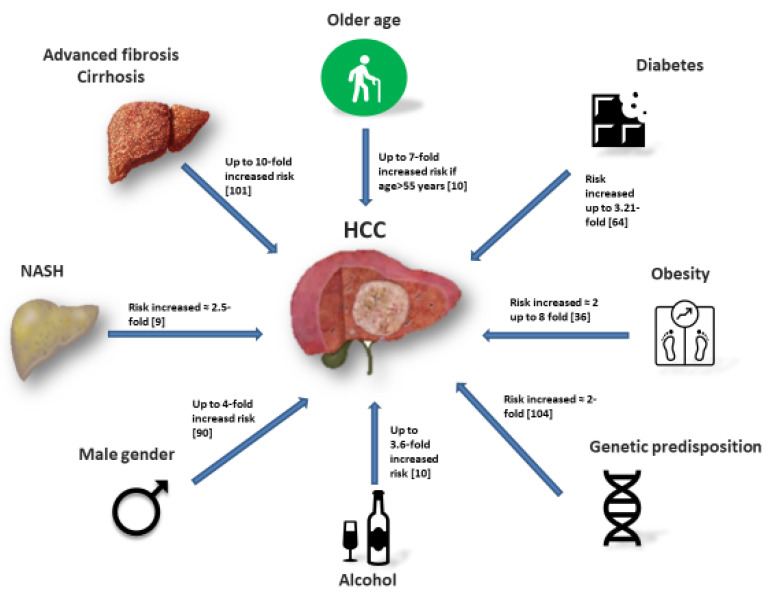
Main risk factors for hepatocellular carcinoma (HCC) among NAFLD patients.

**Table 1 cancers-12-01422-t001:** HCC incidence and main risk factors among the NAFLD population.

First Author (Country, Year)	Type of Study	Number and Type of Patients	Diagnostic Method for NAFLD/NASH	NAFLD Patients with Cirrhosis	Mean Follow Up	HCC Incidence	HCC-Independent Risk Factors (HR, 95% CI) among NAFLD (Multivariate Analysis)
**Ascha (US, 2010) [37]**	Retrospective cohort study	195 NASH-cirrhosis 315 HCV-cirrhosis	Histology or cryptogenic cirrhosis and MetS	100%	3.2 years	NASH: 2.6%	Any alcohol consumption (HR 3.6 (1.6–8.9)) Older age (HR 1.08 (1.02–1.1))
**Yatsuji (Japan, 2009) [39]**	Prospective cohort study, observational	68 NASH-cirrhosis 69 HCV-cirrhosis	Histology	100%	NR	NASH: 5-year occurrence rate = 11.3%	NA
**Kanwal (US, 2018) [9]**	Retrospective cohort study	296,707 NAFLD 296,707 matched controls	Elevated ALT and exclusion of other etiologies of liver disease	0.4% at baseline	9 years	NAFLD: 0.08 per 1000 person-years (PY) Subgroup analyses: -NAFLD + diabetes = 0.45 per 1000 PY -NAFLD + age > 65 = 0.41 per 1000 PY -NAFLD + age > 65 + Hispanic ethnicity = 0.93 per 1000 PY NASH-cirrhosis: 10.6 per 1000 PY (range 1.6–23, highest in older (>65 years) Hispanics). If cirrhosis + high FIB-4 = 13.55 per 1000 PY (11.93–15.33)	Cirrhosis Age ≥ 65 years Hispanic ethnicity Diabetes Male sex Among cirrhosis, risk highest if: -Male sex -Hispanic ethnicity and age ≥ 65 years -Diabetes-FIB-4 score > 2.67
**Ioannou (US, 2019) [54]**	Retrospective cohort study	7068 NAFLD-cirrhosis 16,175 ALD-cirrhosis	If comorbid with diabetes or BMI > 30	100%	3.7 years	Annual incidence = 1.56% If FIB-4 > 3.65, annual incidence = 2.68%	Older age (aHR ≈ 2.09 if age > 60) Male sex (Ahr = 1 versus 0.25 for female) Platelet count < 150 × 10^3^ µL (aHR ≈ 2 to ≈3) Albumin < 3.7 g/dL (aHR ≈ 2 to ≈ 3) AST/ALT ratio > 8.8 (aHR ≈ 2 to ≈ 5)
**Yang (US, 2020) [55]**	Retrospective cohort study	354 NASH-cirrhosis	Histology or history of steatosis or fatty liver at radiology	100%	47 months	5-year cumulative incidence rate = 7.8%	Older age (per decade, HR = 1.8 (1.2–2.6)) Low albumin (HR 2.1 (1.5–2.9)) Diabetes (HR 4.2 (1.2–14.2))
**Yasui (Japan, 2011) [51]**	Cross-sectional multicenter study	87 NASH-related HCC cases	Histology	51%	NR	NR	Advanced fibrosis (21%) and cirrhosis (51%), male sex (62%) and diabetes (59%), obesity (62%) and hypertension (55%) were highly prevalent in the population. Risk analysis was not performed.
**Piscaglia (Italy, 2016) [13]**	Multicenter observational prospective study	145 NAFLD- related HCC cases 611 HCV-related HCC cases	Histology or radiology	53.8%	NR	NR	Causality not assessed but in comparison with the HCV cohort, NAFLD patients showed significantly higher prevalence of male gender, diabetes, hypertension and dyslipidemia
**Kawamura (Japan, 2011) [56]**	Retrospective cohort study	6508 NAFLD patients	Ultrasound scan Only 16 patients had NAFLD-related HCC	NR	5.6 years	Overall incidence = 0.25% Annual incidence = 0.043% Cumulative HCC incidence: -4-year = 0.02% -8-year = 0.19% -12-year = 0.51%	AST ≥ 40 IU/L (HR 8.20 (2.56–26.26)) Age > 60 (HR 4.27 (1.30–14.01)) Platelet count < 150 × 10^3^/µL (HR 7.19 (2.26–23.26)) Diabetes (HR 3.21 (1.09–9.50)) APRI > 1.5 (i.e., significant fibrosis) (HR 25.03 (9.02–69.52))
**Lee (Taiwan, 2017) [11]**	Population-based retrospective cohort study	18,080 NAFLD patients	Not reported	NR	6.3 years	Cumulative incidence at 1-year = 0.18%, increasing until up to 2.73% at 10 years	Age > 55 years (HR 7.78 (3.12–19.44)) ALT elevation (HR 6.80 (3.00–15.42)) 10-year cumulative incidence 4-fold higher in patients aged over 55 with ALT elevation
**Tokushige (Japan, 2013) [57]**	Prospective cohort study	14,530 HCC cases: 84.1% viral aetiology; 7.2% alcoholic, 2% NAFLD; 5.1% cryptogenic	Histology	62%	NR	5-years incidence = 11.3%	Older age (HR 1.103 (1.050–1.159) + Male gender (HR 4.680 (1.803–12.146)) Advanced liver fibrosis (HR 2.718 (1.745–4.233)) Higher GGT (HR 1.005 (1.001–1.009))
**Liu (UK, Switzerland, 2014) [58]**	Prospective cohort study	100 NAFLD-related HCC cases 275 NAFLD cases w/o HCC	Histology or radiology	Among NAFLD-HCCs, 67% Among NAFLD w/o HCC, 26%	NR	NR	Carriage of the PNPLA3 rs-738409 G > C polymorphism (2.26 (1.23–4.14)) Male gender (HR 11.11 (4.17–33.33)) Age (HR 1.24 (1.17–1.32)) Cirrhosis (HR 9.37 (3.82–23.00))
**Grimaudo (Italy, 2020) [59]**	Prospective cohort study	471 NAFLD cases	Histology or radiology	11.5%	64.6 months	Incidence rate in the non-cirrhotic vs. cirrhotic: - 1-year: 0.2% vs. 1.3% - 5 -years: 3.0% vs. 9.3% - 10-years: 4.2% vs. 13.5%	Advanced fibrosis and cirrhosis (HR not reported) PNPLA3 G variant (HR 2.68 (1.01–7.26)). Among the subgroup of patients with F3–F4 fibrosis it was the only independent risk factor: HR 2.66 (1.02–7.13)

NAFLD, non-alcoholic fatty liver disease; NASH, non-alcoholic steato-hepatitis; ALD, alcoholic liver disease; HCV, hepatitis C virus; HCC, hepatocellular carcinoma; NR, not reported; PY, person-years; GGT, gamma-glutamyl transpeptidase; ALT, alanine aminotransferase; AST, aspartate aminotransferase; MetS, metabolic syndrome; HR, hazard ratio; aHR, adjusted hazard ratio; CI, confidence interval.

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
