# Peer review of "Surveillance for Hepatocellular Carcinoma in Patients with Non-Alcoholic Fatty Liver Disease: Universal or Selective?"

_cancers, 2020, doi:10.3390/cancers12061422_

Round 1

Reviewer 1 Report

Non-Alcoholic Fatty Liver Diseases ( NAFLD) have overtaken chronic hepatitis- C- virus infection as the most chronic liver disease in the western world in recent years. NAFLDs cover a broad spectrum of liver diseases, ranking from non-alcoholic-fatty liver (NAFL), non-alcoholic fatty liver hepatitis ( NASH), and secondary fatty liver to fatty cirrhosis. These include insulin resistance, central obsity, genetic and environmental factors, and changes in intestinal microbiota. The sharp increase in NAFLD, especially NASH, is closely related to the pandemic increase in metabolic multisystemic diseases. NASH has a multifactorial genesis, in which, in addition to genetic and lyfestile factors ( super-and malnutrition leading to excessive fat accumulation), mitochondrial dsyfunctions, endotoxins, pro-inflammatory cytokines contribute to chronic hepatocyte inflammation. NASH is per se considered a risk factor for the development of cirrhosis or hepatocellular carcinoma ( HCC). NASH can lead to the development of HCC even in the absence of significant liver fibrosis, especially in the presence of metabolic syndrom. All measures to monitor patient risk for the development of HCC are of great undividual and social importance.

Following a rather extensive introduction, some of which seeming too lenghtly ( e.g. on the costeffectiveness of monitoring strategies to be selected), a presentation of the global significance of the occurrence of HCC (2) in NASH is given 3 chapters, a separate presentation of the data on the occurrence in NAFLD with cirrhosis (3) and NAFLD without cirrhosis (4). Both with and without cirrhosis, NASH patients are considered a risk group for development of HCC and therefore require controlled regular monitoring to detect and treat any eveloping HCC at the earliest time. Particular attention must be paid to cirrhosis patients with metabolic syndrome. Depending on the findings, these examinations should be carried out at 3 or 6-month intervals in combination with clinical examination, laboratory and imaging procedures. Similarly, patients with NASH without cirrhosis or high fibrosis but showing metabolic comorbidities should be screened at least once in a year. In the following chapters the importance of additional factors in non-cirrhotic NAFLD patients, such as diabetes (6), obesity (7), demographic factors (8; predominance of male persons concerned: male-female ratio of 3:1.. 9:1), genetic risk factors (9) and influence of lifestyle (10;  alcohol, smoking) are discussed. Finally, in chapter 11 the methods and procedures for monitoring patients are presented and commented on, while there are no specific recommendations for the screening of NAFLD patients so far. In addition to regular clinical examination, a sonographic evaluation by experienced examiner, also CT or MRI diagnostics where necessary, the APF should be determined, although only 70% informative value, in the absence of other applicable methods, as well as the liver parameters ( ASAT; ALAT; AP; Bili, GGT). New biochemical markers for the detection of HCC should be tested for their practicability.

The thesis given a very good overview of the current state of knowledge regarding the relation between  a NAFLD and an HCC identifies options for monitoring. Explanations are factual and well documented by extensive evidence (125 references). The literature data, on which this document is built, are correct and reproduced with comprehensive expertise. The work is written in academically correct English. Figures and tables provide a good complement to the text. 

Author Response

Please, see attachment.

Reviewer 2 Report

Torres et al. review the most recent evidence on the epidemiology and risk factors (ex. age, sex, ethnicity, diabetes, obesity, genetic predisposition, alcohol consumption and other lifestyle factors) for HCC in patients with NAFLD, with and without cirrhosis, and the evidence supporting surveillance for early HCC detection in these patients. The authors concluded that it is currently difficult to propose general recommendations for HCC surveillance strategies, and best clinical judgement should be exercised, based on the profile of risk factors specific to each patient. This article is well written, but some specific points should be addressed

Major points

1. The authors mentioned that Hispanics were more likely to develop NAFLD-related HCC (page 13). How about Asian population? The authors should add the discussion regarding incidence of NAFLD-related HCC in Asian population, which was reviewed in the following article.

Li J, Zou B, Yeo YH, et al. Prevalence, incidence, and outcome of non-alcoholic fatty liver disease in Asia, 1999-2019: a systematic review and meta-analysis. Lancet Gastroenterol Hepatol. 2019 May;4(5):389-398.

2. Kanwal et al. [8] discussed that FIB-4 can be easily applied in clinical practice to identify the at-risk groups for targeted evaluation and risk modification among the masses of individuals with NAFLD, because an FIB-4 score >2.67 was associated with a high HCC risk in their study. How could the authors discuss the utility of FIB-4 for HCC surveillance strategies.

Minor point

1. There are some typos (ex. line 93, he potential limitations>the potential limitations). Please check.

2. In Table 1, the publication years are incorrect in some references. Please check. What does ‘/’ mean in this Table? In Page 10, what does ‘cryptogenic’ mean in this Table?

Author Response

Please, see attachment
